# Teachers’ Punishment Intensity and Student Observer Trust: A Moderated Mediation Model

**DOI:** 10.3390/bs14060471

**Published:** 2024-06-01

**Authors:** Zhen Zhang, Chunhui Qi

**Affiliations:** 1Faculty of Education, Henan Normal University, Xinxiang 453007, China; zhangzhenpsy@126.com; 2Faculty of Education, Henan University, Kaifeng 475001, China

**Keywords:** punishment intensity, observer trust, trustworthiness, group relationship

## Abstract

During social interactions, people decide whether to trust an actor based on their punitive behaviour. Several empirical studies have indicated that punishment intensity impacts observer trust, yet the underlying mechanism remains to be elucidated. This study included 242 junior high school students and was conducted to investigate the relationship between teachers’ punishment intensity and levels of student bystander trust. Additionally, the mediating role of trustworthiness and the moderating role of group relationships were explored. The results showed that the relationship between punishment intensity and observer trust follows an inverted U-shaped pattern. In addition, mild punishment boosts observer trust by improving perceived trustworthiness (ability and integrity) compared to no punishment, while harsh punishment reduces observer trust more than mild punishment by diminishing perceived trustworthiness (ability, benevolence, and integrity). More importantly, group relationships positively moderate the relationship between punishment intensity and observer trust. Specifically, compared to mild or no punishment, harsh punishment decreases trustworthiness (ability, benevolence, and integrity) in close teacher–student relationships but has less impact on neutral relationships. The above findings demonstrate that guiding educators in developing appropriate disciplinary concepts contributes to enhancing student observer trust.

## 1. Introduction

As the lubricant of social systems, interpersonal trust refers to the willingness of people to put social resources at the disposal of others and take risks when they lack sufficient information to judge their motives and behaviours [1]. The cultivation of strong social relationships hinges on the presence of interpersonal trust, which can eliminate people’s worries, promote cooperative behaviour, and boost social and economic growth [2,3]. Scholars from various disciplines have examined the occurrence, development mechanisms, and influencing factors of interpersonal trust through the utilisation of various assessment tasks, such as self-report questionnaires and economic game theory paradigms [4,5,6,7]. Several studies have indicated that individuals tend to exhibit a predisposition towards trust [8], a trait that is partially influenced by genetic factors [9] and further shaped by the interplay of situational and personality variables [10,11].

During social interactions, individuals assess the trustworthiness of others by considering a range of cues, including personal characteristics (such as facial expressions and vocal intonation) and overt behaviours (such as acts of assistance or punishment), ultimately determining whether to place their trust in them [12,13]. Recent research indicates that third-party punishment intensity may impact observers’ trustworthiness evaluations and trust behaviours toward the punisher [1,14]. However, the current research predominantly concentrates on adult populations and organisational contexts, neglecting the examination of adolescent cohorts within the realm of educational administration. Moreover, during adolescence, which is typically defined as a period spanning from 10 to 24 years of age [15], individuals experience heightened affective sensitivity and an elevated level of social awareness [16]. Consequently, adolescents, whether in the role of perpetrators or observers, exhibit heightened concern regarding the impact of social behaviour on their own or others’ social impressions [17]. Therefore, it is imperative to investigate the impact of teachers’ punishment intensity in school management scenarios on observers’ interpersonal trust in said teachers.

### 1.1. Punishment Intensity and Observer Trust

Managers in organisational management are adept at using rewards and punishments (carrots and sticks) to foster desirable behaviour and deter problematic conduct, particularly within the realm of education [5]. Punishment not only has a direct influence on the individuals responsible and those affected but also has ripple effects on observers. Punishment intensity is the observer’s perception of the punishment intensity for the violator [18]. The spectrum of punishment intensity in the field of education ranges from mild verbal warnings to extreme expulsion from school. Just deserts theory (JDT) indicates that punishment intensity should be commensurate with the level of transgression; thus, appropriate sanctions foster trust, whereas inappropriate penalties (either excessive or lenient) erode trust [19]. Some studies have examined the impact of punishment intensity on the association between punishment and observer trust in organisational contexts and found that observer trust is increased by appropriate sanctions but eroded by inappropriate penalties [1,18]. Accordingly, punishment intensity can influence observer trust among adolescents, showing that the relationship between punishment intensity and observer trust follows an inverted U-shaped pattern (Hypothesis 1).

### 1.2. Trustworthiness as a Potential Mediator

Trustworthiness pertains to an individual’s propensity to fulfil the favourable expectations of others regarding a specific behaviour, serving as a proximal antecedent variable of trust [20]. The construct of trustworthiness encompasses three underlying dimensions: ability, benevolence, and integrity. Multiple research studies have demonstrated that ability, benevolence, and integrity are strong predictors of trust in leaders, teachers, or managers for individuals [1,21,22]. The costly signalling theory (CST) indicates that punishment serves as a demonstrative indicator to observers regarding the trustworthiness and reputation of the sanctioner [23]. Nevertheless, the signalling effect of punishment is not consistent and can be influenced by the intensity of the punishment. Trustworthiness, as perceived by the observer, tends to increase within a specific range as punishment becomes more intense [14,24]. However, once a certain threshold is surpassed, excessive discipline can diminish the trustworthiness of the disciplinarian [1,18]. Additionally, the relationship between punishment intensity and observer trust is mediated by self-reported trustworthiness [1]. Hence, we propose that inappropriate punishment, whether excessive or lenient, can undermine the observer’s trust by diminishing their perception of trustworthiness; conversely, appropriate punishment can bolster the observer’s trust by augmenting their perception of trustworthiness. (Hypothesis 2).

### 1.3. Group Relationship as a Potential Moderator

Group relationships constitute a crucial component of an individual’s self-concept, which emanates from the individual’s perception of collective identity and their emotional investment in group dynamics [25,26]. Social identity theory (SIT) posits that heightened awareness of group relationships in intergroup contexts can lead to in-group favouritism and out-group discrimination, resulting in the preferential allocation of resources and positive evaluations to in-group members and the withholding of resources and negative evaluations towards out-group members [27]. This inclination towards favouring the in-group results in adults and children displaying increased tolerance towards in-group violators [28], issuing lenient punishments to said violators [29,30], and viewing this leniency or protection as an inherent norm [31] or a special obligation [32]. For example, Sun et al. [33] found that punishment administered by out-group members could foster observer trust by bolstering perceptions of integrity, whereas punishment meted out by in-group members could diminish observer trust by diminishing perceptions of benevolence. Consequently, bystanders anticipate that in-group transgressors will exhibit greater leniency toward wrongdoers than regular relationships; thus, excessive punishment from in-group members undermines observers’ trust by diminishing their perception of trustworthiness (H3).

In brief, the aim of this study was to examine the underlying mechanisms that elucidate the connection between the intensity of punishment and observer trust among adolescents through the use of a moderated mediation model. A graphical depiction of the research model is provided in Figure 1.

## 2. Method

### 2.1. Experimental Design and Participant

A complete between-subject design of 3 (punishment intensity: none, mild, severe) × 2 (group relationship: neutral, close) was adopted in this study, with 6 different scenarios. The scenarios described a school management violation for which the offending student should receive a mild punishment. Punishments were divided into mild and severe intensities if they were imposed. Moreover, to examine the impacts of group relationships, we manipulated the group relationships between punishers and offenders by dividing them into ordinary relationships and kin relationships. The scenarios were pretested with ten middle school students to ensure their validity and involvement.

Based on an a priori power analysis, the sample size was estimated using G*Power 3.1 [34]. F tests and ANOVA (fixed effects, special effects, main effects, and interactions) in G*Power (version 3.1.9.7) were selected. To detect a medium effect (f^2^ = 0.25), *N* = 158 participants (27 participants per group) with 0.80 power and 0.05 Type I error rates were needed. A junior high school in Henan Province, China, was selected as the experimental site for this study. Six classes were randomly selected for a class-by-class group test. Two hundred and eighty-four questionnaires were distributed. After the removal of missing values or ineffective responses, there were 242 questionnaires with a minimum of 34 valid responses for each scenario. The average age of the participants was 14.11 ± 1.00 years; 52.5% were female, and 95.5% were not only children.

A trained research assistant distributed and collected situational questionnaires, and standardised processes were followed for completing them. The data collection process was voluntary and anonymous to avoid anticipatory or response bias. Participants were informed that all the data would be used solely for scientific purposes and would not be linked to individual evaluations or academic achievement. Any subject who felt uncomfortable during the study was allowed to withdraw. Health problems were not present, and all subjects gave their informed consent verbally. Completion of the experiment took approximately 20 min. Ethics committee approval was obtained from the Faculty of Education at Henan Normal University, and protocol adherence to the Declaration of Helsinki was ensured.

### 2.2. Experimental Procedure and Materials

Using a pen-and-paper test, six classes were randomly assigned to one of six scenarios in which students acted as bystanders and rated their social perceptions of the teacher who imposed the punishment. The test consisted of four parts: basic demographic variables, hypothetical punishment scenarios, manipulation checks, and social evaluations of the punisher. The test was always administered in the same order as shown below.

#### 2.2.1. Basic Demographic Variables

To control for any differences between individuals, participants were asked to report their age, gender, and whether they were children.

#### 2.2.2. Hypothetical Punishment Scenario

Based on previous research [1], in a hypothetical school punishment scenario, subjects, as bystanders, observed the punishment imposed on violators by a teacher surnamed Sun. In this hypothetical case, the participant (the subject) and Xiao Ming (the violator) were classmates, and Sun was the teacher. Xiao Ming was described as a kin (close distance) or an ordinary student (neutral distance) of Teacher Sun. In the last semester’s final math exam, Xiao Ming was deemed to be absent because he was 30 min late. Sun, as an inspection teacher, followed the school management rules and gave Xiao Ming disciplinary probation (severe punishment), a warning (mild punishment), or did nothing (no punishment). We asked the subjects to rate Sun’s trustworthiness and their interpersonal trust in Sun. All responses were scored on a 5-point scale.

#### 2.2.3. Manipulation Checks

The subjects were asked to answer three manipulation check questions to assess their understanding of the intensity and appropriateness of punishment and the social relationships between Xiao Ming and Sun. The first question inquired about the intensity of the punishment Xiao Ming received, that is, how severe was the punishment Sun imposed on Xiao Ming? The second question asked about the appropriateness of the punishment Xiao Ming received, that is, how appropriate the punishment Sun imposed on Xiao Ming was. The third question asked about the relationship between Sun and Xiao Ming, that is, how close Sun and Xiao Ming were.

#### 2.2.4. Social Evaluations of the Punisher

We selected nine items from Mayer and Davis [35] to assess subjects’ perceptions of Teacher Sun’s ability, benevolence, and integrity. We used Mplus 8.3 to conduct confirmatory factor analysis of the three-factor model to explore the rationality of the model. The results showed that χ^2^/df = 3.82, RMSEA = 0.08, CFI = 0.94, TLI = 0.92, and SRMR = 0.05, indicating an acceptable fit of the model. The subjects’ evaluations of Sun were reliable for ability, benevolence, and integrity (α = 0.69, 0.85, 0.88). The ability, benevolence, and integrity indices were calculated by averaging each respondent’s responses. Moreover, based on previous research [36,37], three behavioural orientation questions were used to measure subjects’ trust in Teacher Sun (e.g., how much would you like to share a secret with Sun?; α = 0.89). Confirmatory factor analysis (CFA) for the trust construct found a fully saturated model, so the ft indices were not reported.

### 2.3. Data Analysis

Statistical analysis was performed using SPSS 26.0 to compute descriptive statistics and conduct correlation analyses and analyses of covariance, with demographic variables included as covariates. In order to examine conditional indirect relationships, path analysis was conducted within a regression-based framework employing Mplus 8.3 [38]. Statistical analyses were performed by conducting 5000 bootstrap resamplings to determine the 95% confidence intervals for indirect effects. Prior to analysis, the study variables were standardised in order to improve the interpretability of the findings.

## 3. Results 

### 3.1. Manipulation Check

Both manipulations were successful. Based on an independent sample *t* test of social closeness scores, subjects rated the group relationship between Sun and Xiao Ming as closer when they were kin (*M* = 2.64, *SE* = 0.12) than when they had a normal teacher–student relationship (*M* = 1.64, *SE* = 0.08), *t*(240) = 7.00, *p* < 0.001. As shown by the one-way ANOVA on the evaluation of punishment severity and appropriateness, the main effects of punishment intensity and appropriateness were significant, *F*s(2, 239) > 17.60, *p*s < 0.001, partial η^2^ > 0.13. Subjects rated disciplinary probation (*M* = 3.55, *SE* = 0.15) as more severe than warning (*M* = 3.07, *SE* = 0.14), *p* < 0.05, the latter of which they rated as more severe than no punishment (*M* = 2.30, *SE* = 0.15), *p* < 0.001. Moreover, subjects rated warnings (*M* = 3.52, *SE* = 0.14) as more appropriate than warnings (*M* = 2.34, *SE* = 0.15) and no punishment (*M* = 2.47, *SE* = 0.15), *p*s < 0.05, while there was no significant difference between the latter two, *p* > 0.05.

### 3.2. Preliminary Analyses

A 3 (punishment intensity) × 2 (group relationship) MANOVA on ability, benevolence, integrity, and trust revealed that the main effects of punishment intensity were significant. *F*s(2, 236) > 6.48, *p*s < 0.01, partial η^2^s > 0.04. Mild punishment led to more ability, benevolence, integrity, and trust than severe punishment, *p*s < 0.01, and more ability and integrity than no punishment, *p*s < 0.01; no punishment also led to more trust than severe punishment, *p* < 0.05. Moreover, the interactions between punishment intensity and group relationships were significant for benevolence, integrity, and trust. *F*s(2, 236) > 3.65, *p*s < 0.05, partial η^2^s > 0.03, and marginally significant for ability, *F*(2, 236) = 2.43, *p* = 0.09, partial η^2^s = 0.02. Pairwise comparisons indicated that when the teacher–student relationship was neutral, mild punishment only induced higher ability ratings than no punishment and severe punishment, *p*s < 0.05, but the latter two had no significant difference, *p*s > 0.05. In contrast, when the teacher–student relationship was kin, mild punishment induced higher ability and integrity ratings than no punishment, *p*s < 0.01, and greater ability, benevolence, integrity, and trust than severe punishment, *p*s < 0.01, and severe punishment caused lower ability, benevolence, integrity, and trust than no punishment, *p*s < 0.05. In terms of punishment, contrasts between close and neutral distances revealed no significant effects of mild or no punishment on ability, benevolence, integrity, or trust (*p* > 0.05). In contrast, when Xiao Ming was severely punished, the effects of group relationships on ability, benevolence, integrity, and trust were significant, with ability, benevolence, integrity, and trust being lower when severe punishment is made by a kin teacher than when it is made by a neutral teacher (*p* < 0.05). The main effects of group relationships were not significant (*p* > 0.05). The means of these effects are shown in Figure 2.

The descriptive statistics and correlations of the variables are reported in Table 1. Punishment intensity was significantly negatively associated with trust (*r* = −0.15, *p* < 0.05). Ability was significantly positively associated with benevolence, integrity, and trust (*r* = 0.59, *p* < 0.01; *r* = 0.67, *p* < 0.01; *r* = 0.61, *p* < 0.01). Benevolence was significantly positively correlated with integrity and trust (*r* = 0.63, *p* < 0.01; *r* = 0.68, *p* < 0.01). Integrity was significantly positively correlated with trust (*r* = 0.70, *p* < 0.01). Furthermore, an independent samples *t* test also showed that gender did not have a significant influence on perceived unfairness, anger, or third-party punishment (*t*s < 1.67, *p*s > 0.05).

### 3.3. Moderated Mediation Model

To assess the impact of punishment intensity on the trustworthiness-mediated process mechanism of trust and its boundary conditions, a moderated mediation model was constructed using Mplus 8.3, with punishment intensity decision as a multicategorical independent variable; trustworthiness as mediators, and group relationships as moderators. This model revealed acceptable fit indices: χ^2^/df = 1.40, RMSEA = 0.04, CFI = 0.99, TLI = 0.98, and SRMR = 0.03.

First, Helmert coding was used to compare both punishments against no punishment (see Table 2). The results for D1 (punishment versus no punishment) showed that D1 and D1 × W (group relationship) could not significantly predict trustworthiness or trust. In contrast, for D2 (severe punishment versus mild punishment), severe punishment (versus mild punishment) weakened trustworthiness (*β* = −0.61, *p* < 0.01), which in turn decreased trust (*β* = 1.12, *p* < 0.01). Moreover, the interaction between D2 and group relationships had a significant effect on trustworthiness (*β* = 0.41, *p* < 0.01), which indicated the influence of punishment (severe punishment versus mild punishment) on trust via trustworthiness depending on the group relationship (see Table 3). More specifically, this negative indirect effect of severe punishment on trustworthiness holds only when the punishment is made by a kin teacher (*β* = −1.15, [−1.53 −0.78]) and not when it is made by a neutral teacher (*β* = −0.20, [−0.57 0.15]). Figure 3a presents path diagrams for this model.

Second, the same interaction analysis using indicator coding was conducted to compare mild or severe punishment against no punishment (see Table 2). The results for D1 (mild punishment versus no punishment) revealed that mild punishment (versus no punishment) enhanced trustworthiness (*β* = 0.40, *p* < 0.01), which in turn increased trust (*β* = 1.12, *p* < 0.01). The results for D2 (severe punishment versus no punishment) revealed that the interaction between D2 and group relationship had a significant effect on trustworthiness (*β* = 0.34, *p* < 0.05), indicating that the effect of punishment (severe punishment versus no punishment) on trust via trustworthiness was dependent on the group relationships (see Table 3). Specifically, this negative indirect effect of severe punishment on trustworthiness was significant only when the punishment was made by a kin teacher (*β* = −0.60, [−0.99 −0.25]) and not when the punishment was made by a neutral teacher (*β* = 0.15, [−0.30 0.60]). Figure 3b presents path diagrams for this model.

## 4. Discussion

The current study explored the relationship between punishment intensity and observer trust among adolescents and its potential mechanisms. The findings demonstrate a curvilinear relationship between punishment intensity and the perception of trustworthiness and trust, characterized by an inverted U-shaped pattern, especially for the close teacher–student relationship condition. Furthermore, compared to the normal relationship condition, moderate punishment (as opposed to no punishment) can significantly enhance perceived ability and integrity, thereby bolstering the observer’s trust in the relative condition. Conversely, severe punishment (in contrast to moderate punishment) can considerably undermine perceived ability, benevolence, and integrity, consequently diminishing observer trust. 

### 4.1. Punishment Intensity and Adolescent Observer Trust

Consistent with the first hypothesis, the relationship between punishment intensity and observer trust follows an inverted U-shaped pattern. Specifically, when people acted as observers, the mild punishment or no punishment imposed by teachers on deviant students resulted in stronger trust ratings than severe punishment, but there was no significant difference between the former two. This result aligns with the fundamental principle of the just deserts theory [19,39], which dictates that punishment should be commensurate with the offence; otherwise, it may give rise to inquiries and distrust. A large body of research based on organisational management also supports the finding that inappropriately harsh punishment by leaders undermines bystander trust [1,18]. The spillover effect of punishment recurrence in adolescents not only expands the research scope from organisational management to educational management but also broadens the age range from adults to adolescents. In other words, akin to adults or employees within organisations, adolescent students are equally concerned with the impartiality of teacher discipline and adjust their trust in disciplinarians accordingly.

### 4.2. The Mediating Role of Trustworthiness

Similar to the pattern observed with trust, the trustworthiness (i.e., ability, benevolence, and integrity) of observers also follows an inverted U-shaped curve as the intensity of punishment increases. In line with the perspectives of costly signal theory [23] and just deserts theory [39], these findings indicate that disciplining consistent with transgression communicates reliable and high-quality signals of trustworthiness, thereby bolstering bystanders’ perceptions of ability, benevolence, and integrity. In addition, we found that trustworthiness mediates the relationship between punishment intensity and observer trust among junior high school students. In particular, mild punishment was found to positively influence the observer’s perception of trustworthiness, resulting in increased trust compared to no punishment. Consistent with previous research [14,24], as the intensity of punishment increases within a certain range, observers tend to perceive the individual as more trustworthy and to elicit greater levels of trust. Conversely, severe punishment was found to have a detrimental effect on the observer’s perception of trustworthiness compared to mild punishment, ultimately leading to a decrease in trust, which further substantiates the findings of earlier studies [1]. Furthermore, our results align with prior scholarly investigations that have demonstrated the role of trustworthiness (i.e., benevolence and integrity) as a mediator in the association between punishment and trust in the context of organisational management [1]. Thus, teachers’ appropriate use of discipline can enhance their own trustworthiness and promote the trust of onlookers, while excessive discipline can diminish their trustworthiness and weaken the trust of onlookers.

### 4.3. The Moderating Effect of the Group Relationship

The present investigation also revealed the moderating effect of the group relationship on the association between punishment intensity and observer trust among adolescents, thus supporting Hypothesis 3. Specifically, the stringent penalties enforced by kin teachers significantly reduced bystanders’ ratings of ability, benevolence, and integrity compared to mild or no punishment. Conversely, severe punishments administered by regular teachers led to diminished effects on trustworthiness or, in some cases, no negative consequences. As posited by social identity theory, individuals tend to display a pronounced inclination toward their in-group [40], viewing it as a norm or obligation [31,32]. Consequently, disproportionate penalisation from kin teachers might erode the trust of observers by diminishing their trustworthiness. In line with our finding, Sun et al. [33] found that punishments from out-group members increased trust by promoting trustworthiness, while punishments from in-group members decreased trust by weakening trustworthiness. Therefore, in the context of a relative teacher–student relationship, the mediating role of trustworthiness in the relationship between excessive discipline and trust is more prominent than that in traditional teacher–student dynamics.

## 5. Educational Suggestions

The present research not only reveals the cognitive process by which discipline intensity affects observer trust (specifically, the mediating role of trustworthiness) but also clarifies the potential boundary conditions (specifically, the moderating role of group relations). These results offer valuable insights into the impact of disciplinary intensity on bystander students’ trust in disciplinarian educators among adolescents. Additionally, this study highlights the indirect pathways to trustworthiness that are most salient in close teacher–student relationships. Consequently, advocating for and providing training in the concept of appropriate disciplinary intensity is crucial for fostering a rational and peaceful educational environment.

Foremost, school administrators at all levels must prioritise the cultivation and promotion of appropriate disciplinary concepts. This goal can be achieved through the dissemination of rules and regulations, showcasing various disciplinary misconduct cases, and sharing daily management experiences with teachers to ensure a clear understanding and reasonable application of punishment intensity. Additionally, educators should enhance their knowledge of educational theory, update their educational concepts, and carefully consider the rationality and appropriateness of disciplinary methods in actual class management to prevent excessive disciplinary intensity. Finally, when dealing with deviant behaviour exhibited by students within their in-group (such as children of relatives or friends), teachers should exercise caution to avoid causing a crisis of trust due to excessive punishment.

## 6. Limitations and Future Research

Similar to previous studies, this research has several limitations that should be acknowledged. Initially, transgressions committed by middle school students spanned multiple domains, such as disruption of classroom dynamics, misbehaviour, and bullying. While existing studies primarily focus on missing test scenarios, further investigation is warranted to validate the findings in alternative contexts. Furthermore, the current research exclusively focuses on junior high school students, which limits its ability to elucidate the origins of the spillover effect resulting from punishment. Some studies have indicated that children aged 5–6 are capable of inferring an actor’s moral motivation based on their behaviour [41], suggesting that future research could delve into the developmental trajectory of the spillover effect of punishment. Additionally, the concept of pedagogical punishment is a heavily context-sensitive concept. In some parts of the world, the mere talk of punishment in a pedagogical context would be considered illegitimate, while in other parts of the world it may be acceptable. Such cultural and regional differences may limit the general applicability of the current findings. Finally, we utilised natural cues, specifically relatives, to examine group relationships, which can be manipulated through artificial cues, natural cues, or social cues. Prior research has indicated that different cues can result in differing degrees of group identification [42]. Consequently, forthcoming studies will investigate the impact of group relationships formed by artificial and social cues on the association between punishment intensity and bystander trust.

## 7. Conclusions

In the current study, we examined the impact of appropriate punishment on bolstering bystander trust and excessive punishment on undermining bystander trust from the perspective of punishment proportionality. Mild punishment boosts observer trust by improving perceived trustworthiness (ability and integrity) compared to no punishment, while harsh punishment reduces observer trust more than mild punishment by diminishing perceived trustworthiness (ability, benevolence, and integrity). Finally, the indirect impacts of trustworthiness are more pronounced in close teacher–student relationships than in neutral teacher–student relationships.

## Figures and Tables

**Figure 1 behavsci-14-00471-f001:**
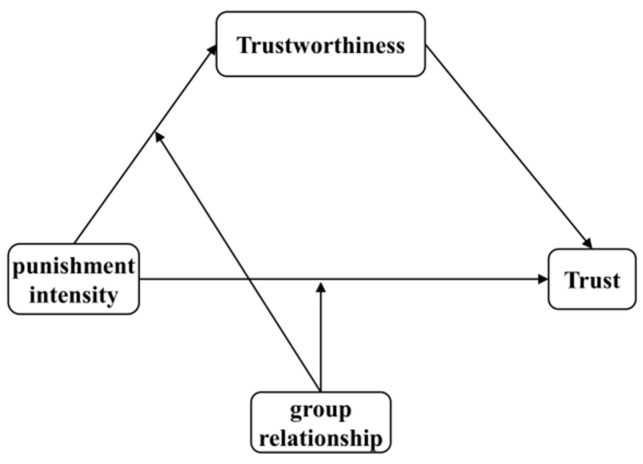
Research model.

**Figure 2 behavsci-14-00471-f002:**
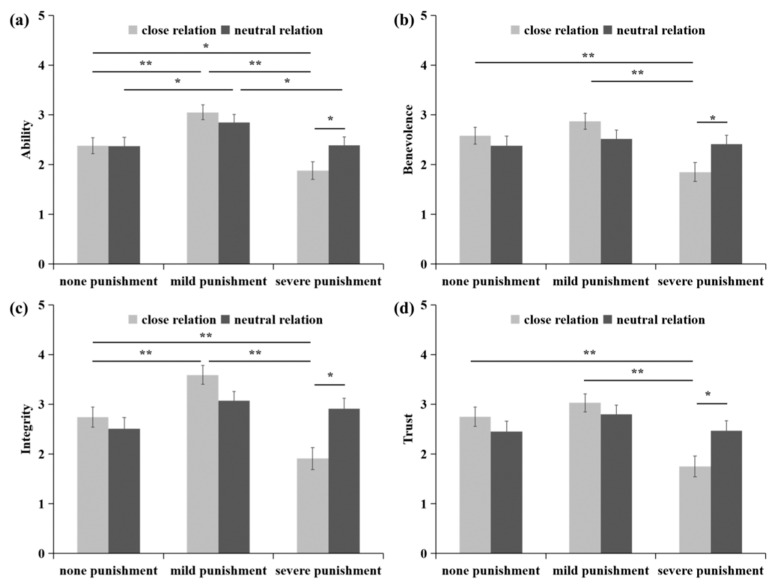
The effects of punishment on ratings of ability (**a**), benevolence (**b**), integrity (**c**), and trust (**d**); * *p* < 0.05, ** *p* < 0.01.

**Figure 3 behavsci-14-00471-f003:**
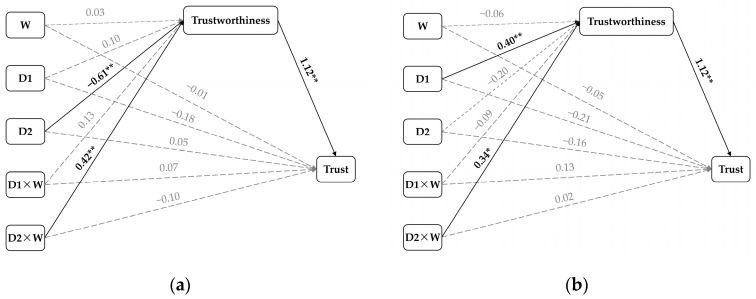
Mplus diagram for moderated mediation. (**a**) Testing whether trustworthiness mediates the effects of punishment versus no punishment (D1) or severe punishment versus mild punishment (D2) on trust and the moderating of group relationships (W). (**b**) Testing whether trustworthiness mediates the effects of mild punishment versus no punishment (D1) or severe punishment versus no punishment (D2) on trust and the moderating of group relationships (W). All the regression coefficients are standardised, * *p* < 0.05, ** *p* < 0.01.

**Table 1 behavsci-14-00471-t001:** Descriptive statistics and correlations of variables (*N* = 242).

Variable	*M*	*SD*	1	2	3	4	5
1. Punishment intensity	0.98	0.79	-				
2. Group relationship	0.49	0.50	0.07	-			
3. Ability	2.53	1.10	−0.08	0.03	-		
4. Benevolence	2.47	1.15	−0.11	−0.02	0.59 **	-	
5. Integrity	2.84	1.39	−0.05	0.01	0.67 **	0.63 **	-
6. Trust	2.59	1.28	−0.15 *	0.01	0.61 **	0.68 **	0.70 **

Note: no punishment = 0; mild punishment = 1; severe punishment = 2; close distance = 0; neutral distance = 1; * *p* < 0.05, ** *p* < 0.01.

**Table 2 behavsci-14-00471-t002:** Mplus results for the moderated mediation model (*N* = 242).

Regression Equation		Helmet Coding ^a^			Indicator Coding ^b^		
Result Variable	Prediction Variable	*β*	*t*	95% CI	*β*	*t*	95% CI
Trustworthiness	D1	0.10	0.84	[−0.14 0.34]	0.40	2.85 **	[0.13 0.69]
	D2	−0.61	−4.80 **	[−0.87 −0.38]	−0.20	−1.57	[−0.47 0.05]
	W	0.03	0.50	[−0.07 0.13]	−0.06	−0.57	[−0.26 0.14]
	D1 × W	0.13	1.07	[−0.09 0.37]	−0.09	−0.66	[−0.34 0.18]
	D2 × W	0.42	3.75 **	[0.21 0.65]	0.34	2.57 *	[0.09 0.61]
Trust	D1	−0.18	−1.88	[−0.38 0.01]	−0.21	−1.90	[−0.42 0.01]
	D2	0.05	0.47	[−0.14 0.25]	−0.16	−1.46	[−0.37 0.06]
	W	−0.01	−0.07	[−0.09 0.07]	−0.05	−0.59	[−0.24 0.12]
	D1 × W	0.07	0.74	[−0.12 0.28]	0.13	1.16	[−0.09 0.34]
	D2 × W	−0.10	−1.01	[−0.30 0.10]	0.02	0.20	[−0.21 0.25]
	Trustworthiness	1.12	10.45 **	[0.92 1.33]	1.12	10.47 **	[0.92 1.33]

Note: Dummy coding of multicategorical independent variables: ^a^ D1 = no punishment (−0.66), mild punishment (0.33), and severe punishment (0.33); D2 = no punishment (0), mild punishment (−0.50), and severe punishment (0.50). ^b^ D1 = no punishment (0), mild punishment (1), and severe punishment (0); D2 = no punishment (0), mild punishment (1), and severe punishment (1). * *p* < 0.05, ** *p* < 0.01.

**Table 3 behavsci-14-00471-t003:** Conditional indirect effect of group relationships on trust (*N* = 242).

Mediator Variable	Prediction Variable	Moderator Variable	Helmet Coding ^a^			Indicator Coding ^b^		
			*β*	*t*	95% CI	*β*	*t*	95% CI
Trustworthiness	D1	close	−0.03	−0.17	[−0.36 0.29]	0.55	2.85	[0.18 0.93]
	D1	neutral	0.25	1.20	[−0.17 0.66]	0.35	1.55	[−0.09 0.79]
	D2	close	−1.15	−6.02 **	[−1.53 −0.78]	−0.60	−3.25 **	[−0.99 −0.25]
	D2	neutral	−0.20	−1.12	[−0.57 0.15]	0.15	0.66	[−0.30 0.60]

Note: Dummy coding of multicategorical independent variables: ^a^ D1 = no punishment (−0.66), mild punishment (0.33), and severe punishment (0.33); D2 = no punishment (0), mild punishment (−0.50), and severe punishment (0.50). ^b^ D1 = no punishment (0), mild punishment (1), and severe punishment (0); D2 = no punishment (0), mild punishment (1), and severe punishment (1). ** *p* < 0.01.

## Data Availability

The data presented in this study are available on request from the corresponding author.

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
