# Peer review of "Teachers’ Punishment Intensity and Student Observer Trust: A Moderated Mediation Model"

_behavsci, 2024, doi:10.3390/bs14060471_

Round 1

Reviewer 1 Report

Comments and Suggestions for Authors

The relationships presented in the model suggest that research could be better developed with structural equations.

Although the authors present another statistical technique, which may also be acceptable. However, the research has several methodological deficiencies:

--The authors do not specify whether the constructs are made up of reflective or formative indicators.

--The authors also do not present an evaluation of the various types of validity (for example, convergent, discriminant...). And they do not indicate the level of reliability of the scales used in the research. This last deficiency is critical, since the replicability of the study cannot be established. And therefore, the results are only anecdotal or contextualized to the data used in the research.

As a consequence, the results cannot be extrapolated and the scientific methodology of the study is questioned.

Author Response

Response to review 1

Manuscript ID: behavsci-2994511

Title: Teachers’ punishment intensity and student observer trust: A moderated mediation model

Thanks so much for your works on the manuscript. Based on the comments of reviewer 1, we have carefully addressed the issues raised in the review comments, point by point, and made appropriate changes (using blue text) in the manuscript.

Q1. The relationships presented in the model suggest that research could be better developed with structural equations.

Replay: Thanks for this comment. As suggested, we used the structural equation model with Mplus (Byrne, 2016) to conduct confirmatory factor analysis for the trustworthiness construct and re-examined the moderated mediating effect model.

References:

Byrne, B.M. Structural Equation Modeling with Amos: Basic Concepts, Applications, and Programming, 3rd ed.; Multivariate applications series; Routledge, Taylor & Francis Group: New York, NY, USA, 2016; ISBN 978-1-138-79702-4.

Q2. The authors do not specify whether the constructs are made up of reflective or formative indicators.

Replay: Thank you for pointing to these important issues. The experimental materials used in this study are all adapted or selected from previous studies. Specifically, the hypothetical punishment scenario was adapted from the study by Wang et al. (2017); the trustworthiness scale was selected from Mayer and Davis (1999); the trust scale was adapted from previous studies (Liberman and Shaw, 2018; Keri et al., 2009).

Moreover, confirmatory factor analysis (CFA) revealed acceptable fit indices for the trustworthiness construct: χ2/df = 3.82, RMSEA = 0.08, CFI = 0.94, TLI = 0.92, SRMR = 0.05. Confirmatory factor analysis (CFA) for the trust construct found a fully saturated model, so the ft indices were not reported.

References:

Keri, S.; Kiss, I., Kelemen, O. Sharing secrets: oxytocin and trust in schizophrenia. Social Neurosci. 2009, 4, 287-293. https://doi.org/10.1080/17470910802319710

Liberman, Z.; Shaw, A. Secret to friendship: Children make inferences about friendship based on secret sharing. Dev. Psychol. 2018, 54, 2139–2151. https://doi.org/10.1037/dev0000603

Mayer, R. C.; Davis, J. H. The effect of the performance appraisal system on trust for management: A field quasi-experiment. J. Appl. Psychol. 1999, 84, 123–136. https://doi.org/10.1037/0021-9010.84.1.123

Wang, L.; Murnighan, J. K. The dynamics of punishment and trust. J. Appl. Psychol. 2017, 102, 1385–1402. https://doi.org/10.1037/apl0000178

Q3. The authors also do not present an evaluation of the various types of validity (for example, convergent, discriminant...). And they do not indicate the level of reliability of the scales used in the research. This last deficiency is critical, since the replicability of the study cannot be established. And therefore, the results are only anecdotal or contextualized to the data used in the research.

Replay: Thanks for this comment. We ensure the validity and repeatability of the study from the following aspects.

First of all, just as the reply to question 2, the experimental materials in the current study are adapted or selected from previous studies, which ensures the validity of the study to a certain extent.

Secondly, confirmatory factor analysis (CFA) for the trustworthiness construct found acceptable fit indices. Moreover, the internal consistency coefficients of ability, benevolence, integrity and trust are 0.69, 0.85, 0.88, and 0.89, respectively. These results indicate that the trustworthiness and trust scales have strong reliability and validity.

Finally, we re-analyze the moderated mediation model with structural equation model with Mplus, and the results are in agreement with those based on the Process plug-in. Moreover, This model revealed acceptable fit indices: χ2/df = 1.40, RMSEA = 0.04, CFI = 0.99, TLI = 0.98, SRMR = 0.03.

Therefore, we consider the current findings to be reliable and valid. 

Please also see the attachment.

Reviewer 2 Report

Comments and Suggestions for Authors

In many ways, the paper is well written, and it fulfills academic standard and expectations. Concerning relevance of references, research design and argumentation, I am very positive. Particularly, the research model presented in figure 1 is very good. Also, the complete between-subject design of 3 (punishment intensity: none, mild, severe) × 2 (group relationship: neutral, close) is worthy of recognition, and the case described is fine. Finally, from a technical point of view the analyses of the results are very convincing.

In contrast, the discussion part is debatable, because it is based on the conceptualization of punishment as a quantifiable concept. Also, I am not convinced by the section concerning educational suggestions, because I miss specific pedagogical arguments (p. 10). Finally, I am critical to the conclusions section (p. 10), because it is based on a linear distinction between at one end “mild punishment” and at the other end “harsh punishment”. In relation to all these sections I have some fundamental concerns, which are also critically relevant in relation to the basic assumptions of the paper.

My fundamental concerns are rooted in the following two considerations:

First, I have a critical consideration concerning the concept of “teachers' punishment intensity”. My consideration is that I don’t believe that - at least in a pedagogical context - punishment can be graduated into mild vs. harsh punishment. There are qualitative differences between what you call “mild” punishment, e.g. verbal abuse, and what you call “harsh” punishment, e.g. physical punishment or other sanctions. If a teacher reacts to e.g. an essay by saying that this isn’t good enough and points to the specific weaknesses in the essay, the verbal abuse would be part of a critical argument and thus could be considered a legitimate and constructive verbal abuse, while e.g. physical punishment isn’t an integrated part of an academic reasoning and thus would be considered illegitimate. Thus, I cannot accept the fundamental assumption of a quantitative intensity line between mild and harsh punishment. You have to make qualitative distinctions.

Let me provide an example: On p. 2 you make a distinction between “appropriate sanctions” and “inappropriate penalties”. But whether a sanction or a punishment is appropriate or inappropriate in a teaching context is not a question of mild vs. harsh, but it is a pedagogical question: if the sanction is part of a pedagogical intervention, e.g. an academic argumentation, it is appropriate. If the punishment is a result of human anger or irritation, it is inappropriate.

Another example can be found in the discussion section on p. 9: Here, it says that “punishment should be commensurate with the offense; otherwise, it may give rise to inquiries and distrust”. This is a classical principle in legal theory and practice. However, I doubt that it can be applied directly in a pedagogical context. Here, one must always consider whether the type of punishment is appropriate in relation to the learning and development of the punished person. Again, one has to use qualitative considerations, not just scalable “degrees” of punishment.

My second consideration is that the concept of pedagogical punishment is a heavily context sensitive concept. In some parts of the world, e.g. Northern Europe, where I come from, the mere talk of punishment in a pedagogical context would be considered illegitimate, while in other parts of the world it may be acceptable.

Author Response

Response to review 2

Manuscript ID: behavsci-2994511

Title: Teachers’ punishment intensity and student observer trust: A moderated mediation model

Thanks so much for your works on the manuscript. Based on the comments of reviewer 2, we have carefully addressed the issues raised in the review comments, point by point, and made appropriate changes (using blue text) in the manuscript. 

Q1. I have a critical consideration concerning the concept of “teachers' punishment intensity”. My consideration is that I don’t believe that - at least in a pedagogical context - punishment can be graduated into mild vs. harsh punishment. There are qualitative differences between what you call “mild” punishment, e.g. verbal abuse, and what you call “harsh” punishment, e.g. physical punishment or other sanctions. If a teacher reacts to e.g. an essay by saying that this isn’t good enough and points to the specific weaknesses in the essay, the verbal abuse would be part of a critical argument and thus could be considered a legitimate and constructive verbal abuse, while e.g. physical punishment isn’t an integrated part of an academic reasoning and thus would be considered illegitimate. Thus, I cannot accept the fundamental assumption of a quantitative intensity line between mild and harsh punishment. You have to make qualitative distinctions.

Replay: Many thanks to the reviewers for their constructive comments. Here, we have a candid communication with the reviewers on the definition and distinction of the intensity of educational punishment in this study.

First, as the reviewers point out, corporal punishment or verbal abuse is prohibited by law in most parts of the world, including China, where our authors are from, so the punishments mentioned in the current study are legal. Specifically, Chinese law stipulates that students can be criticized by name, offered an apology, and self-examination orally or in writing for minor violations. Students who seriously violate the rules may be disciplined, suspended from school, or placed on probation. In short, there are quantitative distinctions and differences in educational punishment in the eastern cultural background.

Second, we also agree with the reviewer's point of view that the type of punishment is appropriate in relation to the learning and development of the punished person. In fact, the appropriateness of educational discipline is determined by the match between the student's transgression and the teacher's punishment. Among them, the former is often affected by factors such as harm, subjective intention and number of violations (Jordan et al., 2016; Krueger and Hoffman, 2016; Salcedo et al., 2024; Wang et al., 2017; Yang et al., 2024), while the latter is affected by factors such as the size, type and motivation of punishment (Anderson et al., 2020; Dorison et al., 2020; Levine et al., 2018; Li et al., 2023; Maeda et al., 2022; Sun et al., 2023; ). In other words, from the perspective of experimental manipulation, there are a large number of operable variables, forms, and spaces for disciplinary appropriateness. The current study only selects the match between disciplinary intensity and the degree of violation to manipulate disciplinary appropriateness.

Third, It is true that the emotional motivation and educational purpose of teachers to implement discipline as described by the reviewers will certainly affect the effect of discipline. The current study assumes a moderate transgression scenario and examines trust spillover effects of excessive or light discipline solely in terms of discipline intensity by manipulating teachers' discipline intensity (no, mild, and severe). Therefore, from the perspective of experimental control, the above factors are additional variables that need to be controlled and need to be discussed in future studies.

In short, in the current study, both moderate and severe punishments are legitimate, and their purpose is to promote students' development, rather than punishment caused by teachers' anger. They differ only in the degree of punishment. 

References:

Anderson, R. A., Crockett, M. J., & Pizarro, D. A. (2020). A theory of moral praise. Trends in Cognitive Sciences, 24(9), 694-703. https://doi.org/10.1016/j.tics.2020.06.008

Dorison, C. A., Umphres, C. K., & Lerner, J. S. (2022). Staying the course: Decision makers who escalate commitment are trusted and trustworthy. Journal of Experimental Psychology: General, 151(4), 960–965. https://doi.org/10.1037/xge0001101

Jordan, J. J., Hoffman, M., Bloom, P., & Rand, D. G. (2016). Third-party punishment as a costly signal of trustworthiness. Nature, 530(7591), 473-476. https://doi.org/10.1038/nature16981

Krueger, F., & Hoffman, M. (2016). The emerging neuroscience of third-party punishment. Trends in Neurosciences, 39(8), 499-501. https://doi.org/10.1016/j.tins.2016.06.004

Levine, E. E., Bitterly, T. B., Cohen, T. R., & Schweitzer, M. E. (2018). Who is trustworthy? Predicting trustworthy intentions and behavior. Journal of Personality and Social Psychology, 115(3), 468–494. https://doi.org/10.1037/pspi0000136

Li, Y., Luo, J., Niu, H., & Ye, H. (2023). When punishers might be loved: fourth-party choices and third-party punishment in a delegation game. Theory and Decision, 94(3), 423-465. https://doi.org/10.1007/s11238-022-09897-6

Maeda, K., Kumai, Y., & Hashimoto, H. (2022). Potential influence of decision time on punishment behavior and its evaluation. Frontiers in Psychology, 13, 794953.

Salcedo, J. C., & Jimenez‐Leal, W. (2024). Severity and deservedness determine signalled trustworthiness in third party punishment. British Journal of Social Psychology, 63(1), 453-471. https://doi.org/10.1111/bjso.12687

Shao, B. (2019). Moral anger as a dilemma? An investigation on how leader moral anger influences follower trust. The Leadership Quarterly, 30(3), 365-382. https://doi.org/10.1016/j.leaqua.2018.10.002

Sun, B., Jin, L., Yue, G., & Ren, Z. (2023). Is a punisher always trustworthy? in-group punishment reduces trust. Current Psychology, 42(26), 22965-22975. https://doi.org/10.1007/s12144-022-03395-2

Wang, L., & Murnighan, J. K. (2017). The dynamics of punishment and trust. Journal of Applied Psychology, 102(10), 1385–1402. https://doi.org/10.1037/apl0000178

Yang, Q., Hoffman, M., & Krueger, F. (2023). The science of justice: The neuropsychology of social punishment. Neuroscience & Biobehavioral Reviews, 105525. https://doi.org/10.1016/j.neubiorev.2023.105525

Q2. My second consideration is that the concept of pedagogical punishment is a heavily context sensitive concept. In some parts of the world, e.g. Northern Europe, where I come from, the mere talk of punishment in a pedagogical context would be considered illegitimate, while in other parts of the world it may be acceptable.

Replay: Thank you for pointing to these important issues. As the reviewers have said, pedagogical punishment has great cultural and regional differences. In the East Asian society where the author lives, discipline is a necessary and legitimate means of promoting student development, and is accepted by students, parents, schools, and society. Such cultural and regional differences limit the generalizability and applicability of the current findings. In this regard, we have added this content to the limited section. Please see “Limitations and future research”, as described below.

“Additionally, the concept of pedagogical punishment is a heavily context sensitive concept. In some parts of the world, the mere talk of punishment in a pedagogical context would be considered illegitimate, while in other parts of the world it may be acceptable. Such cultural and regional differences may limit the general applicability of the current findings.”

Of course, to the best of our knowledge, the current study is the first to attempt to explore the spillover effects of punishment intensity on bystander students' trust and its boundary conditions in an educational context. Although the manipulation of variables all referred to previous studies of different disciplines, the tentative and exploratory nature of this study also led to some inadequacies in the experimental design. Thanks again to the reviewers for their valuable and profound comments.

Please also see the attachment.

Round 2

Reviewer 1 Report

Comments and Suggestions for Authors

The authors have corrected the main comments made by the reviewer.

The document is ready to be published.